# A Comparison of the Sensitivity and Cellular Detection Capabilities of Magnetic Particle Imaging and Bioluminescence Imaging

**Sophia Trozzo** [1,2,*], **Bijita Neupane** [1] **and Paula J. Foster** [1,2]

1    Department of Medical Biophysics, Schulich School of Medicine and Dentistry, Western University, London, ON N6A 3K7, Canada; bneupane@uwo.ca (B.N.); pfoster4@uwo.ca (P.J.F.)
2    Robarts Research Institute, Schulich School of Medicine and Dentistry, Western University, London, ON N6A 3K7, Canada
*    Correspondence: strozzo@uwo.ca

**Abstract:** Background: Preclinical cell tracking is enhanced with a multimodal imaging approach. Bioluminescence imaging (BLI) is a highly sensitive optical modality that relies on engineering cells to constitutively express a luciferase gene. Magnetic particle imaging (MPI) is a newer imaging modality that directly detects superparamagnetic iron oxide (SPIO) particles used to label cells. Here, we compare BLI and MPI for imaging cells in vitro and in vivo. Methods: Mouse 4T1 breast carcinoma cells were transduced to express firefly luciferase, labeled with SPIO (ProMag), and imaged as cell samples after subcutaneous injection into mice. Results: For cell samples, the BLI and MPI signals were strongly correlated with cell number. Both modalities presented limitations for imaging cells in vivo. For BLI, weak signal penetration, signal attenuation, and scattering prevented the detection of cells for mice with hair and for cells far from the tissue surface. For MPI, background signals obscured the detection of low cell numbers due to the limited dynamic range, and cell numbers could not be accurately quantified from in vivo images. Conclusions: It is important to understand the shortcomings of these imaging modalities to develop strategies to improve cellular detection sensitivity.

**Keywords:** cell tracking; magnetic particle imaging; bioluminescence imaging; multimodal imaging; sensitivity; cell detection; quantification

## 1. Introduction

Cellular imaging techniques have been used to track cancer cells, stem cells, and immune cells in both preclinical and clinical scenarios [1,2]. In the context of cancer, cell tracking has been helpful in characterizing metastasis [3–6] and for imaging the homing of circulating tumor cells, which have been studied as delivery vehicles for cancer therapeutics [7]. A variety of immune cell types have been tracked. Tumor-associated macrophages have been imaged as a marker of tumor aggressiveness [8]. Multiple modalities have been used to track natural killer cells [9,10], cytotoxic T cells [11,12], and dendritic cells [13] in tumor-bearing mice. Imaging has captured neutrophil infiltration in wound healing [14] and macrophage homing in models of heart disease [15]. In vivo cell tracking has also been employed to monitor the biodistribution of transplanted stem cells [16–18] and for tracking stem cell homing in diseases such as cancer [19] and stroke [20].

Different imaging modalities can be used for cell tracking, each with its own strengths and limitations. The most common modalities include bioluminescence imaging (BLI), positron emission tomography (PET), and magnetic resonance imaging (MRI). Bioluminescence imaging (BLI) is an optical modality that utilizes cells that are genetically engineered to constitutively express a luciferase reporter gene. To generate a signal, a luciferin substrate is administered to animals harboring luciferase-expressing cells. Luciferin is catalyzed by the luciferase enzyme, and photons of light are released as a by-product of this reaction.

Photons are captured by a cooled charge-coupled device camera and converted into an electric signal. BLI is highly specific, as luciferase genes are not endogenously expressed in mammalian cells [21]. It is also highly sensitive, with one group reporting the detection of a single cell in the mouse lung microvasculature [22]. Other major benefits of BLI include the ability to track cell proliferation and viability. There is no label dilution over time, as luciferase is stably expressed through each cell division. Some luciferase enzymes require ATP as a co-factor to drive the enzymatic reaction; hence, only live cells produce signals [21]. However, there are limitations to this modality. BLI signal is attenuated through tissue, and this limits the depth of imaging to 1–2 cm. Luciferase genes and substrates with a further red-shifted, or near-infrared shifted, emission wavelength have improved signal intensity at greater tissue depths [23]. This modality also has no potential for clinical translation, as cells must be engineered to express luciferase [21].

Cells directly labeled with superparamagnetic iron oxide (SPIO) particles can be imaged with magnetic resonance imaging (MRI). SPIOs cause local magnetic field distortions, which leads to rapid dephasing of surrounding protons. This causes signal loss in T2/T2*-weighted images, thus producing a hypointense (dark) region on the image where SPIOs are present [24]. The signal void is larger than the size of a single cell, which provides MRI with single-cell sensitivity under ideal imaging conditions [25]. However, it is not possible to directly quantify cell numbers from these signal voids. SPIO-based MRI also has low specificity, as signal voids produced by SPIOs can be hard to distinguish from other anatomical regions that similarly appear dark [24]. A small number of clinical studies have demonstrated the proof of principle for SPIO-based MRI cell tracking [26–28].

Recently, a new imaging modality has emerged with the capability of directly quantifying SPIO-labeled cells. First reported in 2005, magnetic particle imaging (MPI) detects signals directly from SPIOs. The scanner consists of two electromagnets that generate a gradient magnetic field known as the selection field. At the center of the selection field, there exists a field-free region (FFR) where SPIOs experience net zero magnetization. When an alternating magnetic field known as the drive field is applied, SPIOs within the FFR can respond by flipping their magnetic dipole. SPIOs outside of the FFR are magnetically saturated and cannot respond to the drive field. The FFR is moved across the imaging field of view, and the MPI signal is linearly proportional to the amount of SPIOs within this FFR at any given point. For more information about the physics of magnetic particle imaging, please refer to the review articles by Harvell-Smith et al. [29] and Chandrasekharan et al. [30].

MPI is advantageous over MRI cell tracking, as it provides a directly quantifiable, positive contrast or "hotspot" image of SPIO-labeled cells. The slope of a calibration line measuring the MPI signal of known amounts of SPIO can be used to calculate iron mass from an image. Samples with a known cell number can then be used to calculate the amount of iron per cell, and this value can be used to estimate the number of cells in subsequent images. Direct cell quantification is not possible with any other imaging modality. SPIO-labeled cells can be imaged in vivo, and the MPI signal is not attenuated through tissue [29,30]. However, MPI also does not provide any anatomical information, and a secondary imaging modality is required, such as MRI or CT [4,31]. Table 1 compares BLI, PET, SPIO–MRI, and MPI for preclinical cell tracking.

Multimodal imaging strategies that combine two or more modalities have been widely used to take advantage of the strengths of each imaging modality and minimize the drawbacks [5,13,15,17,19,31,36–38]. This has the potential to provide a more complete picture of the fate of a population of cells in vivo. In this study, we compare MPI to BLI. We also evaluate the strengths and limitations of each modality for in vivo cell detection and discuss the ways that MPI and BLI can be used in a complementary fashion.

**Table 1.** Comparison of BLI, PET, SPIO-based MRI, and MPI for cell tracking in animals.

| | BLI | PET | MRI | MPI |
|---|---|---|---|---|
| **Cell Labeling Agent** | Luciferase genes | Positron-emitting radiotracers | SPIO | SPIO |
| **Source of Signal** | Photons released upon luciferase— luciferin enzymatic reaction [21] | Emission of two gamma rays via annihilation of positron with an electron in tissue [32] | SPIO effects on surrounding protons producing negative contrast on T2/T2* MRI [24] | SPIO response to magnetic fields [30] |
| **Imaging Time** | Seconds–minutes/image (multiple images per sequence) 15–30 min to peak signal (longer in some cases) | 15–30 min (3D) >60 min (longitudinal tracer activity) | 2–10 min (2D) * 10–40 min (3D) * | 2 min (2D) 30 min (3D) |
| **Number of Mice Imaged at Once** | Up to 5 mice ** | Up to 4 mice [33] ** | 1 mouse | 1 mouse |
| **Longitudinal Cell Tracking** | High (>2 months) [12] | Low (<1 week) [32] | Medium (weeks to months) [15,16] | Medium (weeks to months) [4,5] |
| **In Vivo Detection Limits** | Single cell [22] in ideal conditions | 10,000–100,000 cells [34] highly dependent on tracer characteristics | Single cell [25] in ideal conditions | 1000 cells (commercial SPIO) [35] 250 cells [36] (custom SPIO) |
| **Spatial Resolution** | Low (>1 mm) [37] | High (<1 mm) [34] | High (<1 mm) [24] | Low (≥1 mm) [29] *** |
| **Quantification** | Total flux (photons/s) Average radiance (photons/s/cm$^2$/steradian) | Tracer uptake and specific activity | Area or volume of signal loss, degree of signal loss, change in T2/T2* | Iron mass (µg) and cell number |
| **Depth Penetration** | Low (1–2 cm) [23] | No limitation | No limitation | No limitation |
| **Specificity** | High | Medium | Low | High |

* Dependent on imaging sequence; ** dependent on system specifications/setup; *** can be improved with higher gradient strength.

## 2. Materials and Methods

### 2.1. Cell Culture

Mouse 4T1 breast carcinoma cells were cultured in Dulbecco's modified eagle's medium (DMEM) (Wisent Inc., Saint-Jean-Baptiste, QC, Canada) supplemented with 10% fetal bovine serum (FBS) and 5% antimycotic/antibiotic. Cells were passaged every 2–3 days at 80% confluency and maintained at 37 °C and 5% $CO_2$.

### 2.2. Fluorescence Activated Cell Sorting (FACS)

4T1 cells were transduced to co-express firefly luciferase (fluc) and green fluorescent protein (GFP) using the RediFect Red-FLuc-GFP commercial lentiviral vector (Revvity, Waltham, MA, USA). This is the red-shifted *Luciola italica* firefly luciferase, which has a peak emission wavelength of approximately 610 nm [39]. Transduced cells were sorted based on the intensity of GFP fluorescence using a FACSAria III cell sorter (BD Biosciences, Franklin Lakes, NJ, USA) with FACSDiva software (Version 9.0.1, Windows 10) at the London Regional Flow Cytometry Facility (University of Western Ontario, London, ON, Canada). Briefly, cells were suspended in phosphate-buffered saline (PBS) with 2% FBS at a concentration of $30 \times 10^6$ cells/mL. GFP was detected using the blue laser trigon on the B 530/30 channel (blue trigon detector B—530/30 bandpass and 502 longpass filter). Sorted cells were collected in a buffer consisting of 50% DMEM and 50% FBS. After five days in culture, cell viability was assessed based on Sytox blue exclusion. Sytox blue was added to cells at a 1:1000 dilution and was detected on the V 450/40 channel (violet trigon detector C—450/40 bandpass filter). A live cell gate (FSC-A vs. Violet 450/40) was created

to specifically select viable cells. After the sorting, GFP fluorescence was qualitatively assessed using an ECHO 4 revolve microscope (ECHO, San Diego, CA, USA).

### 2.3. Characterization of ProMag

The SPIO particle used in these experiments was a magnetic microsphere called ProMag (ProMag 1 Series—COOH, Bangs Laboratories, IN, USA). ProMag consists of polymer-encapsulated iron oxide particles where many iron nanocrystals are clustered within a polymer matrix. The diameter of each microsphere is 1 μm. The inert polymer coating limits their degradation [40]. The sensitivity and resolution of ProMag were characterized using the RELAX module on a Momentum preclinical MPI scanner (Magnetic Insight, Alameda, CA, USA). Particle sensitivity was defined as the peak amplitude of the point spread function (PSF) normalized to iron content (μg). The resolution of ProMag was measured as the full-width half-maximum (FWHM) of the PSF displaying MPI signal normalized to iron content (μg), and the maximum signal was measured in arbitrary units (A.U.). The FWHM (T) was divided by the gradient field strength used for imaging (3.055 T/m) to determine the resolution in millimeters.

Images of samples of known amounts of ProMag (6.50 μg, 3.25 μg, 1.625 μg, 0.813 μg, and 0.406 μg) were acquired individually by MPI to evaluate the relationship between ProMag concentration and MPI signal. Two-dimensional images (1 projection, 2 min) were acquired using a 12 cm × 6 cm (z, x) field of view (FOV) with high sensitivity settings (3.055 T/m gradient, 20 (x) and 26 (z) excitation amplitudes). An image of the empty sample bed was also acquired for image quantification. A linear regression was used to plot the total MPI signal (A.U.) against iron mass (μg) for each sample. The slope of this line was used to quantify iron mass from subsequent images of cell samples.

### 2.4. Cell Labeling Procedure

Cells were cultured in a T75 cm$^2$ flask supplemented with 10 mL of DMEM. At 80% confluency, cells were labeled with ProMag (25 μg/mL) by introducing 52 μL of ProMag in DMEM (total volume of 10 mL). After a 24 h incubation period, the cells were washed three times with PBS and trypsinized to detach them from the culture flask. Cells were then diluted in DMEM to deactivate the trypsin and centrifuged at 1000 rpm for 5 min. Three additional washing steps were performed in PBS to remove cellular debris and free iron. To isolate labeled cells, magnetic column separation was performed using an EasySep magnet (Stem Cell Technologies, Vancouver, BC, Canada). Cells were placed in a 5 mL flowtube and inserted into the magnet for 5 min. Labeled cells were washed from the sides of the tube with PBS.

### 2.5. Cell Labeling Efficiency—Perls' Prussian Blue and Magnetic Column Separation

Perls' Prussian blue (PPB) staining was performed to visualize ProMag within cells. A total of $2 \times 10^5$ cells suspended in 300 μL of PBS were placed within a cytofunnel attached to filter paper and a glass slide. The apparatus was placed within a cytospin 4 centrifuge (Thermo Fisher Scientific, San Francisco, CA, USA) and centrifuged at 1000 rpm for 5 min. The slide was then fixed in a 3-part methanol, 1-part acetic acid solution for 5 min. A PPB solution was created by dissolving 0.5 g of potassium ferrocyanide in 25 mL of ultra-pure water, followed by the addition of 25 mL of 2% hydrochloric acid. After the solution was filtered, the slide was submerged for 30 min, and iron within the cells was stained blue. Then, the slide was counterstained with a filtered nuclear fast red solution for 5 min, resulting in cell nuclei appearing red. The stained cells were then dehydrated by exposing the slide to increasing concentrations of ethanol (70%, 95%, and 100%), followed by two xylene washes. Finally, the slide was sealed with cytoseal protectant and a coverslip. PPB slides were visualized using the Echo 4 Revolve Microscope (ECHO, San Diego, CA, USA).

To quantify cell labeling efficiency in another way, cell counts were performed before (*n* = 3) and after (*n* = 3) magnetic column separation using the trypan blue exclusion assay (Countess Automated Cell Counter, Invitrogen, Carlsbad, CA, USA). The labeled cells were

prepared following the steps listed above. The % labeling efficiency was quantified by dividing the average cell count after magnetic separation by the average cell count before magnetic separation. This was repeated three separate times, and each time, the % labeling efficiency was quantified.

## 2.6. In Vitro Cell Imaging

### 2.6.1. BLI as a Measure of Cell Viability

The BLI signal of unlabeled and ProMag-labeled cells was compared in vitro to assess the effects of labeling on cell viability. Six replicates of 20,000 cells were added to a black 24-well plate and topped with 500 μL of complete DMEM. The cells were left overnight to adhere to the plate. The following day, half of the samples ($n$ = 3 wells) were labeled with ProMag (25 μg/mL) for 24 h. The next day, all wells (labeled and unlabeled cells) received 5 μL of D-luciferin (30 mg/mL), and the plate was imaged with BLI. After imaging, a media change was performed for each well to remove excess D-luciferin. This process was repeated every day for 5 days after iron-labeling to assess long-term changes in cell viability. BLI images were acquired on an IVIS Lumina XRMS Series 3 In Vivo Imaging System (Revvity, Waltham, MA, USA). The following imaging parameters were used: open filter, auto exposure (1–3 s), 12.5 cm field of view, binning = 8 (high sensitivity), f/stop = 1. Images were acquired until each sample reached the peak radiance ($p/s/cm^2/sr$).

### 2.6.2. Calculating Iron Content per Cell

To estimate the amount of iron present in each cell, triplicate samples of 500,000 ProMag-labeled cells were suspended in 250 μL of DMEM and imaged with MPI. The same scan parameters were used as previously described for imaging the calibration line samples. The amount of iron calculated for these samples was divided by 500,000 (the known number of cells imaged) to calculate iron per cell. The average iron loading per cell was reported as the mean of the three samples imaged.

### 2.6.3. Comparing the Sensitivity of MPI and BLI In Vitro

To directly compare the in vitro sensitivity of MPI and BLI, triplicate samples of ProMag-labeled cells were prepared in a 1:2 dilution series consisting of samples ranging from 51,200 to 100 cells (a total of 30 samples). The samples were suspended in 250 μL of DMEM in small (500 μL) Eppendorf tubes. All samples were imaged on the same day with BLI and MPI, and they were kept on ice when not imaged to preserve cell viability.

Cell samples were imaged with BLI first using the same parameters described in Section 2.6.1. Prior to imaging, 2.5 μL of D-luciferin (30 mg/mL) was added to each tube. One batch of images was acquired for triplicate samples of 51,200 cells to 3200 cells (total of 15 samples, 20 min scan, 2 s exposure time). Then, a second batch of images was acquired for triplicate samples of 1600 cells to 100 cells (total of 15 samples, 20 min scan, 60 s exposure time). Images were acquired until all samples reached peak radiance.

Two-dimensional MPI was performed immediately after BLI. Cell samples were imaged individually using the scan parameters described previously. Individual Eppendorf tubes were placed vertically in the center of the sample holder, positioning them in the center of the MPI field of view. Cell number was calculated by dividing the total iron content (μg) in each image by the average iron per cell. For samples of lower cell numbers that could not be resolved and quantified in 2D, 3D images were acquired (35 projections) with a smaller FOV 3 cm $\times$ 6 cm (z, x) to shorten scan time from 30 min to 17 min.

## 2.7. Mice

In vivo, the sensitivity of MPI and BLI were directly compared using different strains of mice. C57Bl/6 ($n$ = 1), Balb/c ($n$ = 1), and nu/nu ($n$ = 2) mice were obtained from Charles River Laboratories Inc. (Senneville, QC, Canada). All animals were cared for in accordance with the Standards of the Canadian Council on Animal Care and Western University's

Council on Animal Care under an approved animal use protocol. Mice were fasted for 12 h prior to imaging to minimize background gastrointestinal signal in MPI.

### 2.7.1. Cell Preparation for Animal Injection

Cells labeled with ProMag (25 µg/mL) were suspended in 25 µL of PBS and stored on ice prior to each injection. Immediately before injection, each sample was mixed with 25 µL of Matrigel (50 µg/mL) for a combined injection volume of 50 µL. One nu/nu mouse was injected with 6400 labeled cells subcutaneously at the back of the neck. Three other mice (C57Bl/6 ($n$ = 1), Balb/c ($n$ = 1), and nu/nu ($n$ = 1)) were injected with 12,800 labeled cells subcutaneously at the back of the neck. These mice were used to assess the sensitivity and cell detection capabilities of MPI and BLI in different imaging conditions (imaging before versus after hair removal, and prone versus supine). All injections were performed under 2% isoflurane anesthesia in 100% oxygen. Each mouse was imaged with MPI and BLI on the same day.

### 2.7.2. In Vivo MPI

Mice were imaged with MPI first. A full FOV (12 cm × 6 cm) 2D image was acquired for each mouse in the prone position with high-sensitivity scan parameters, as outlined previously. To assess MPI signal at different tissue depths, the Balb/c mouse was imaged in the prone and supine position with hair. The MPI signal from the two images was directly compared. Where signal was detected, the iron content and cell number were also quantified using the methods described previously.

### 2.7.3. In Vivo BLI

Mice were imaged individually with BLI 30–80 min after MPI. Mice received an intraperitoneal injection of 150 µL of D-luciferin (30 mg/mL). Each mouse was imaged for 20–25 min by taking a series of images with a 2 min delay. The following scan parameters were used: open filter, auto exposure (60 s for each mouse), 10 cm field of view, binning = 8 (high sensitivity), f/stop = 1. A representative optical image was taken of each mouse to provide anatomical information. It was also possible to acquire X-ray images using the IVIS system for additional anatomical information. The nu/nu mice (one containing 6400 cells and one containing 12,800 cells) were imaged in the prone position until the peak radiance was reached. The nu/nu mouse containing 12,800 cells was flipped to the supine position, and additional images were acquired to assess the effects of tissue attenuation. The BLI signal in the prone image was compared with the supine image.

For 10 min, the C57Bl/6 and Balb/c mice with 12,800 cells were imaged in the prone position with hair. Then, the image acquisition was briefly paused, the mice were shaved, and Nair was applied to expose the skin around the injection site. This was performed to directly compare the BLI signal with and without hair. The imaging sequence then continued until the peak radiance was reached. We anticipated the peak radiance to arrive 20–25 min after the injection of d-luciferin for each mouse.

### 2.8. Image Analysis

MPI data sets were visualized and analyzed using the Horos™ imaging software. (Version 3 (LGPL-3.0)). Horos is an open-source program that is distributed free of charge under the LGPL license at https://Horosproject.org (accessed on 1 September 2023) and sponsored by Nimble Co LLC d/b/a Purview in Annapolis, MD, USA.

Images of free ProMag and labeled cell samples were analyzed using the standard deviation of an image of the empty sample holder. This standard deviation was multiplied by five. This threshold was applied to select a region of interest (ROI), which was used to quantify the images of free ProMag and labeled cell samples. The total MPI signal (A.U.) was calculated by multiplying the ROI area (mm$^2$) by the mean MPI signal (A.U.).

For in vivo MPI images, the five times standard deviation method was not used to quantify MPI signal. This is because impeding gastrointestinal signals in some images

could also be quantified above this threshold. Instead, a small ROI was drawn around the cell sample in the image, excluding the gut region. The maximum signal from the cell sample was recorded. Then, a new ROI was drawn with a threshold of half the maximum signal, and the area ($mm^2$) and mean signal (A.U.) were multiplied to quantify the total MPI signal (A.U.).

BLI images were analyzed using the Living Image$^{TM}$ software (Revvity, Version 4.7.3.20616). For images of cell samples, an identical ROI was placed over each sample to encircle the region of signal. The radiance ($p/s/cm^2/sr$) was recorded for each ROI.

All mouse BLI images were analyzed using the same scale. On the Living Image$^{TM}$ software, each mouse image was adjusted to the same scale by displaying the same maximum and minimum radiance values. Then, an ROI was drawn to encircle the region of signal for each mouse. For mice where two images were compared (before and after hair removal), an ROI with the same area was used.

*2.9. Statistics*

All statistical analyses were performed using the GraphPad Prism version 10 program. A linear regression was performed to create a calibration line comparing ProMag content with MPI signal, and a constraint was added to force the line through the origin ($x = 0$ and $y = 0$). A two-way ANOVA test was used to assess differences in the BLI signal of ProMag-labeled and unlabeled cell samples. For labeled cell samples, linear regressions were performed to assess the relationship between cell number and BLI signal or MPI signal. To determine the accuracy of cell number calculations for in vitro MPI, a Student's *t*-test with a 95% confidence interval was used to compare the average calculated cell number to the actual cell number for each group of samples: 51,200–3200 cells ($n = 3$ for each cell number, 15 samples total). A correlation plot was created to compare the MPI signal (A.U.) and the BLI signal ($p/s/cm^2/sr$) for each cell sample between 51,200 and 3200 cells ($n = 3$ for each cell number). Statistical significance was defined by *p*-values less than 0.05, and the goodness of fit was determined by the $r^2$ value.

**3. Results**

*3.1. Characterization of 4T1 Fluc Transduction and ProMag Labeling*

4T1 cells were transduced to express fluc, sorted, and then labeled with ProMag for dual BLI and MPI. Cells were sorted based on GFP fluorescent intensity (Figure 1A). A GFP sort gate was created by comparing untransduced 4T1 cells to 4T1 luc GFP+ cells. In total, 87.8% of sorted cells were GFP+. These cells were then expanded in culture for 5 days and resorted with a Sytox blue viability dye to select live and GFP+ cells (Figure 1B). Pre-sort analysis revealed that 96.0% of the total cell population was viable (Sytox blue negative). A post-sort purity analysis revealed that 99.7% of live cells were GFP+. Fluorescent images of sorted cells in culture visually confirmed the high proportion of GFP+ cells (Figure 1C).

A qualitative assessment of labeling efficiency was performed with PPB staining. ProMag was visualized within cells with little presence of unlabeled cells and extracellular iron (Figure 1D). After magnetic column separation, between 5 and 10% of the total cell population was lost, as assessed by cell counting with trypan blue exclusion. Between 90 and 95% of the total cell population was labeled and remained after magnetic column separation (Tables A1–A3, Appendix A). Both methods (PPB and cell counting) showed a high cell labeling efficiency.

The BLI signal from ProMag-labeled and unlabeled cells were compared over a period of five days as a measure of cell viability (Figure 1E). There were no significant differences in average radiance between labeled and unlabeled cells, suggesting no effects of ProMag labeling on cell viability (Figure 1F).

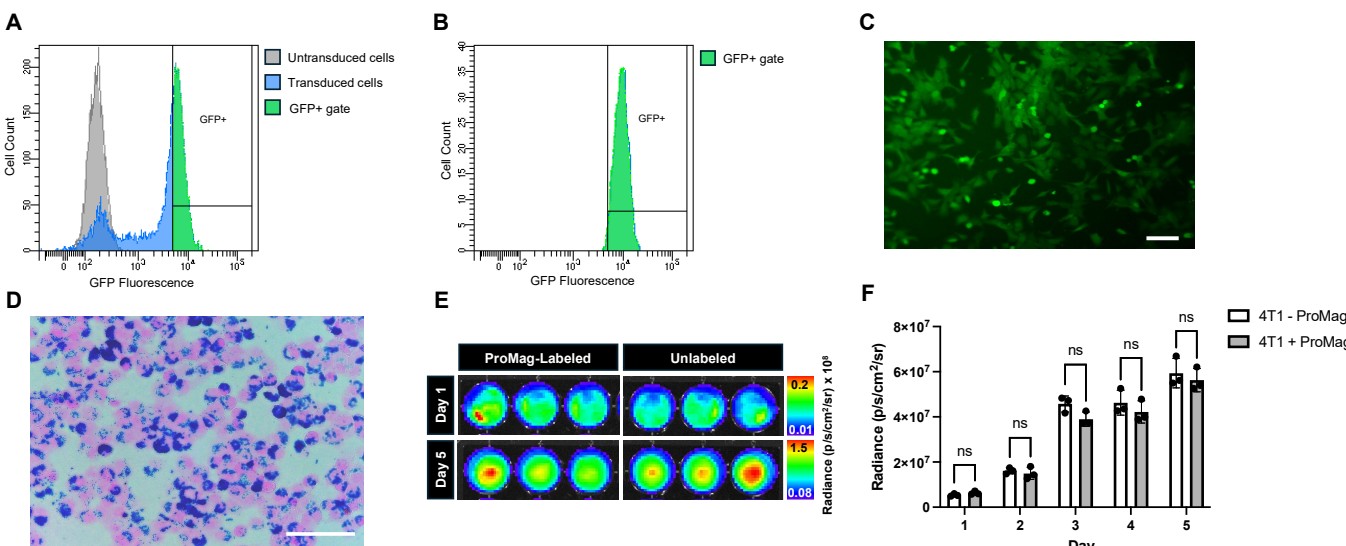

**Figure 1.** Characterization of 4T1 luc GFP+ cell transduction and ProMag-labeling: (**A**) A flow cytometry histogram comparing GFP fluorescence for an untransduced 4T1 cell population (grey) and a transduced 4T1 luc GFP+ cell population (blue). A sort gate was created to select the transduced cells based on GFP fluorescent intensity (green). (**B**) A second flow cytometry histogram showing GFP fluorescence for viable (Sytox blue negative) 4T1 luc GFP+ cells. (**C**) A fluorescent image of sorted 4T1 luc GFP+ cells in culture visually confirms GFP expression (scale bar = 100 μm). (**D**) A Perls' Prussian blue stained image of ProMag-labeled 4T1 luc GFP+ cells shows minimal extracellular iron and high iron-labeling efficiency (scale bar = 100 μm). (**E**) Representative BLI images on day 1 and day 5 post-iron loading of 4T1 cells in a 24-well plate. (**F**) The corresponding radiance measurements for each sample revealed no significant differences between labeled and unlabeled cells up to 5 days post iron loading (ns = non-significant).

### 3.2. Characterization of ProMag

The MPI performance of ProMag was characterized by MPI relaxometry using the scanner's RELAX module. The sensitivity of ProMag was 0.0465 A.U. This value is the peak amplitude of the PSF normalized to iron mass (μg) to allow comparison with other SPIOs (Figure 2A). A second relaxometry curve measuring MPI signal normalized to iron mass and maximum signal was used to determine the resolution of ProMag (Figure 2B). The resolution of ProMag was 7.94 mm, which was determined by dividing the FWHM of the PSF (0.0242 T) by the gradient field strength (3.055 T/m). A calibration line revealed a linear relationship between ProMag iron mass and MPI signal (Figure 2C, $R^2$ = 0.9802, $p < 0.0001$). Figure 2D shows the images of each ProMag sample used for the calibration line, displayed in full dynamic range.

### 3.3. In Vitro Bioluminescence Imaging

Suspended cell samples were first imaged with BLI, then MPI, on the same day (30 samples total). Representative images of 51,200 to 3200 cells (Figure 3A) and 1600 to 100 cells (Figure 3B) show that all cell samples produced BLI signals. Radiance values showed a significant linear and positive correlation with increasing cell number for samples between 51,200 and 3200 cells (Figure 3C, $p < 0.0001$) and 1600 and 100 cells (Figure 3D, $p < 0.0001$).

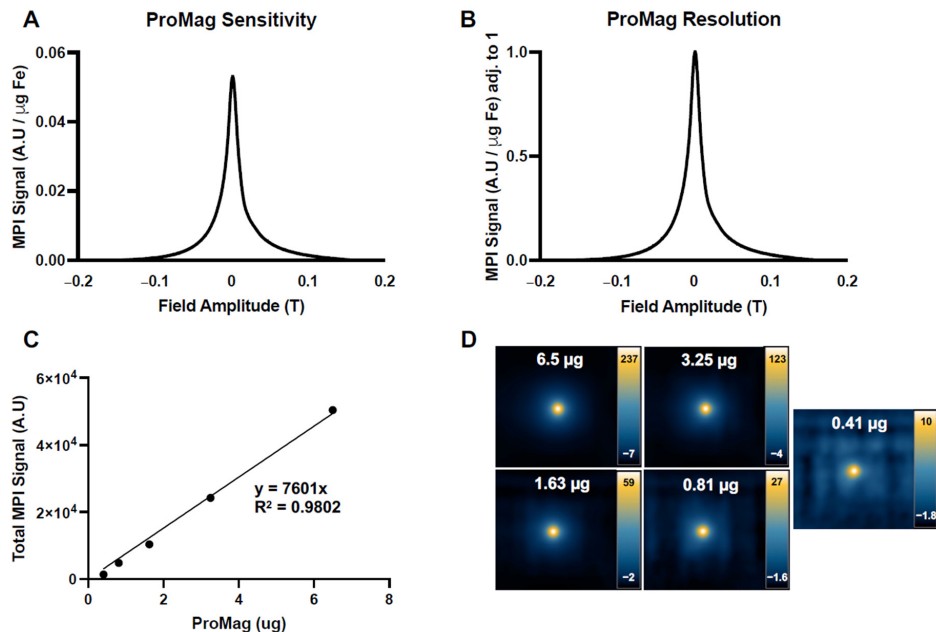

**Figure 2.** Characterization of ProMag: (**A**) A relaxometry curve generated by normalizing MPI signal (A.U.) to iron mass (µg) to measure ProMag sensitivity. (**B**) A second relaxometry curve to measure ProMag resolution. MPI signal was normalized to iron mass and maximum MPI signal. (**C**) A calibration line showing the linear relationship between iron mass and total MPI signal ($R^2$ = 0.9802, *p* < 0.0001). (**D**) Corresponding images of ProMag samples used to generate the calibration line, displayed in full dynamic range. Note the differences in scale.

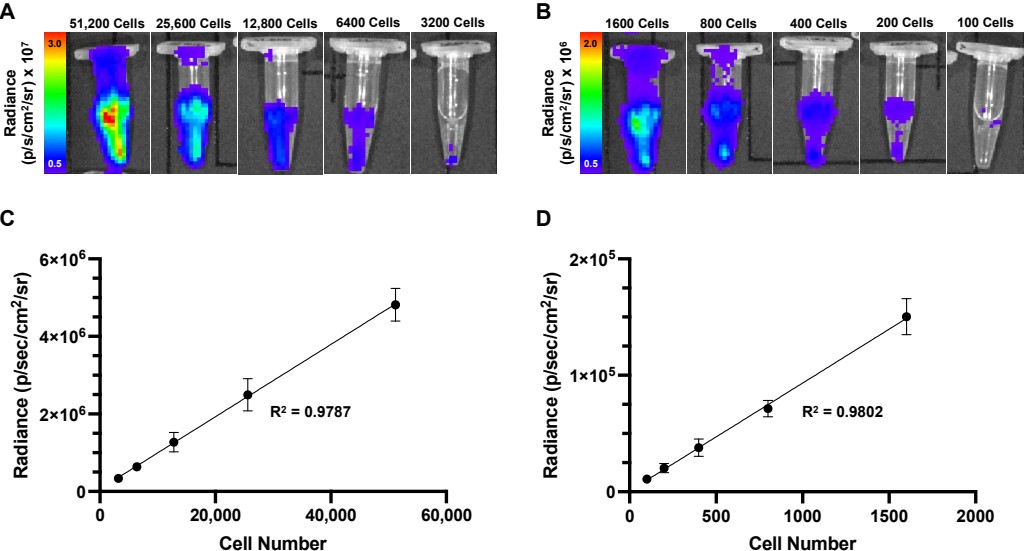

**Figure 3.** In vitro BLI cell detection: (**A**) Representative images showing detectable BLI signal for all cell samples from 51,200 to 3200 cells and (**B**) 1600 to 100 cells. (**C**) A linear regression showing cell number versus radiance measurements for samples from 51,200 to 3200 cells ($R^2$ = 0.9787). (**D**) A linear regression showing cell number versus radiance measurements for samples from 1600 to 100 cells ($R^2$ = 0.9802). Radiance measurements showed a significant linear and positive correlation with increasing cell number for all samples (*p* < 0.0001).

### 3.4. In Vitro Magnetic Particle Imaging

The iron loading per cell was 16.2 ± 0.89 pg. MPI images of 500,000 cell samples used for this calculation are shown in Appendix A (Figure A1). As few as 3200 ProMag-labeled cells (51.9 ng of iron) were detected with a 2D scan. The 1600 cell samples could not be

accurately quantified in 2D due to image noise. Detection limits were improved to as few as 800 cells (13.0 ng of iron) using a 3D scan (Figure 4A). Two-dimensional and 3D images of cell samples were considered detectable if the signal-to-noise ratio exceeded five times the standard deviation of the background noise.

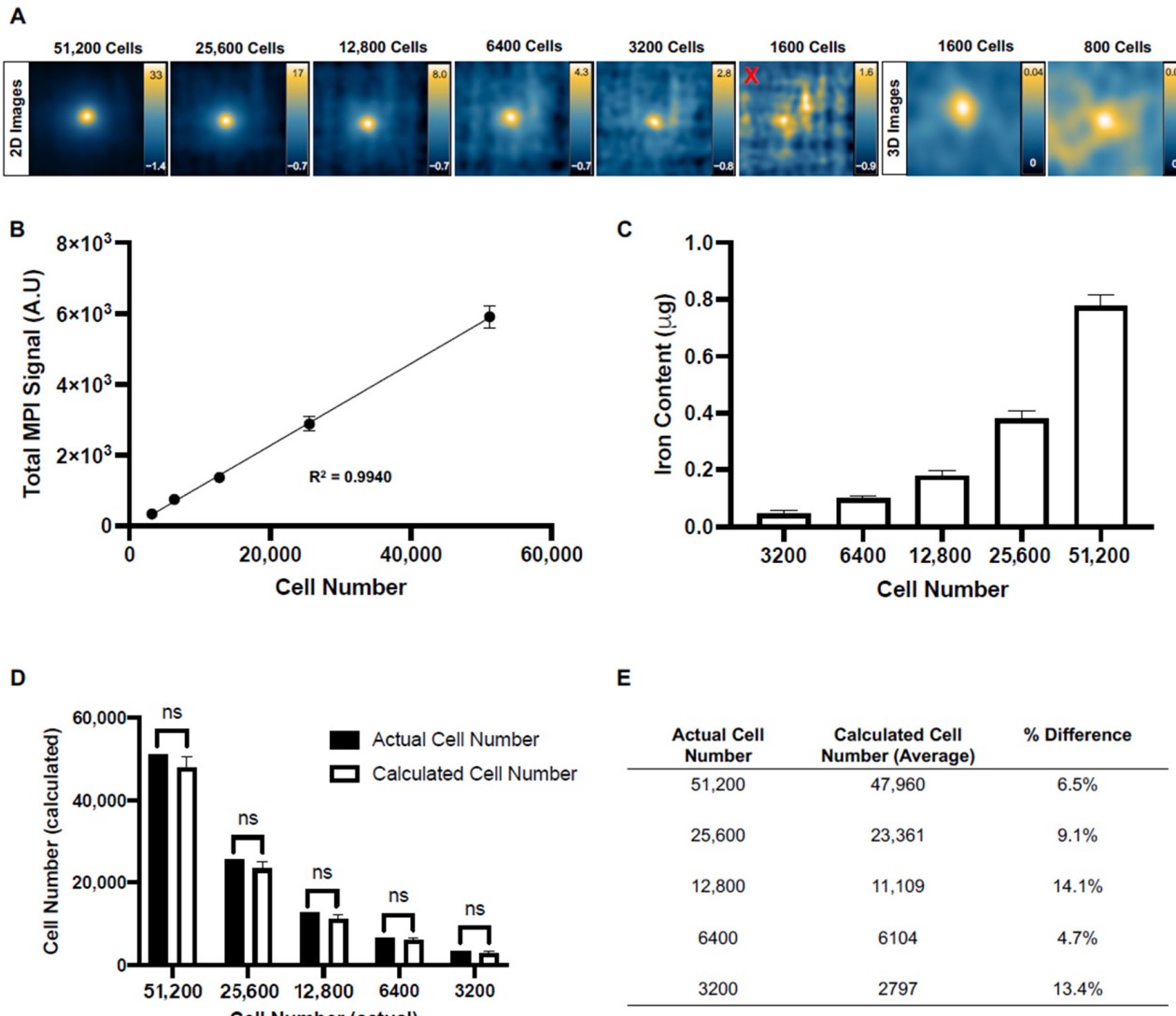

**Figure 4.** In vitro MPI cell detection: (**A**) Representative 2D and 3D MPI images of all cell samples that could be detected and quantified. A red "x" is placed over the 2D image of 1600 cells that could not be accurately quantified due to image noise. (**B**) A linear regression comparing cell number versus total 2D MPI signal (A.U.) for samples of 51,200 to 3200 cells. The total 2D MPI signal showed a significant linear and positive correlation with increasing cell number ($R^2$ = 0.9940, $p < 0.0001$). (**C**) Iron content was calculated for each cell sample that could be quantified in 2D. There was a significant positive and linear correlation between cell number and iron content ($R^2$ = 0.9940, $p < 0.0001$). (**D**) Using the iron content measurements for each sample, the cell number was calculated and compared with the actual cell number. There was no significant difference between the mean calculated cell number and the actual cell number for each group of samples as determined by Student's *t*-tests (ns = non-significant). (**E**) A chart was generated to visualize the % differences between the average calculated cell number and the actual cell number for each group of samples. The largest difference was observed for the 12,800-sample group (14.1%), and the smallest difference was observed for the 6400-cell group (4.70%).

For 2D images, the total MPI signal was linearly and positively correlated with increasing cell number (Figure 4B). Using the slope of a 2D calibration line and the total MPI signal, the iron content of each 2D image was calculated. The mean iron content increased by approximately 2 times as the cell number was doubled (Figure 4C). The iron content of each 2D image was then divided by the average iron loading per cell to obtain a calculated cell number. There were no significant differences between calculated and actual cell numbers, where calculated cell numbers were always underestimated (Figure 4D). The percent difference between the average calculated cell number and the actual cell number was quantified for each group of cell samples imaged with 2D MPI. The highest difference between calculated and actual cell numbers was 14.1% for 12,800 cells, and the lowest difference was 4.70% for the 6400 cells (Figure 4E).

A correlation plot was generated to compare the in vitro BLI signal and total MPI signal of each cell sample between 51,200 and 3200 cells (Figure 5). A linear relationship was observed between the two measurements as cell number increased ($R^2$ = 0.9904, $p < 0.0001$).

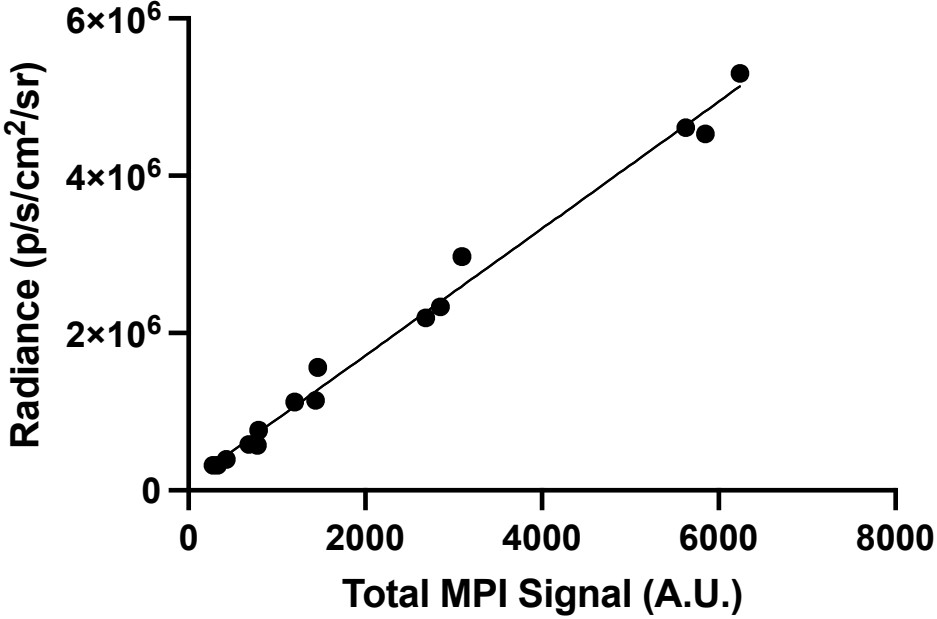

**Figure 5.** Correlation plot comparing total MPI signal (x-axis) and radiance (y-axis) measurements for cell samples between 51,200 and 3200 cells. Each point represents an individual sample (a total of 15 samples). There is a strong linear correlation between MPI and BLI signals for in vitro samples ($R^2$ = 0.9904, $p < 0.0001$).

### 3.5. In Vivo Bioluminescence Imaging and Magnetic Particle Imaging

Two nu/nu mice, one injected with 6400 labeled cells and the other with 12,800 labeled cells, were imaged with MPI and BLI to compare cell detection in vivo (Figure 6). A total of 6400 cells were selected as the lower limit, as it represents one step above the 2D in vitro MPI detection limit of 3200 cells, and we expected lower sensitivity in vivo. In the MPI image, there was a large signal from the gastrointestinal region, which prevented the visualization of the smaller region of signal from the injected cells (Figure 6A). This unwanted signal has been observed before [4,31,41]. The interference with other signals of interest is due to the known low dynamic range in MPI [42]. When this nu/nu mouse was imaged with BLI in the prone position, the signal from 6400 cells was clearly visualized and quantified ($6.38 \times 10^4$ p/s/cm$^2$/sr) (Figure 6B). Another nu/nu mouse was imaged after receiving an injection of 12,800 labeled cells. This cell number was selected to determine if a higher signal from the cells reduced shadowing from the gut signal. Doubling the cell number allowed for visualization and quantification of the signal from these cells in the 2D full FOV MPI image (Figure 6C). BLI signal was also detected for the same mouse ($4.77 \times 10^5$ p/s/cm$^2$/sr) (Figure 6D).

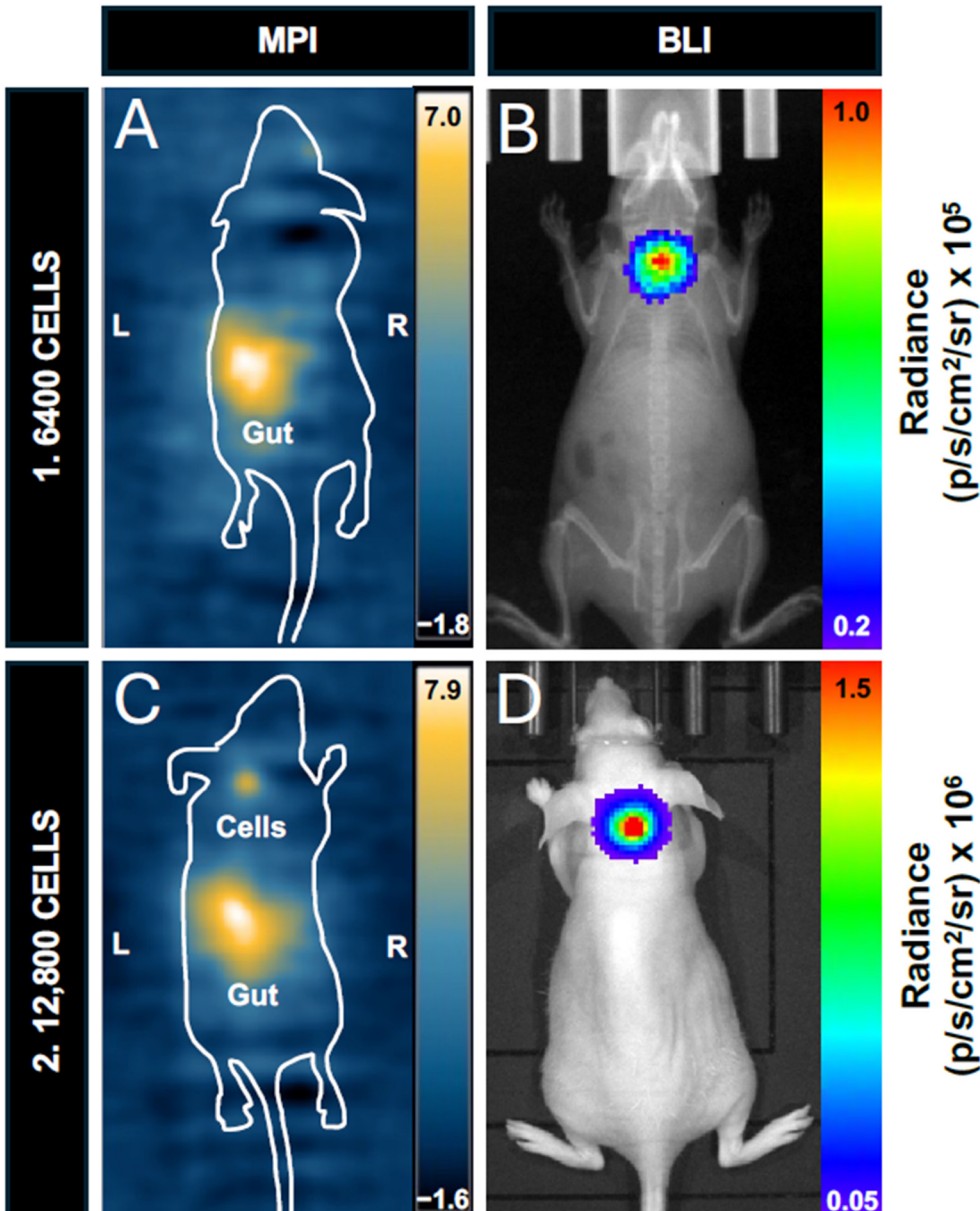

**Figure 6.** Comparing in vivo cell detection in nu/nu mice: (**A**) 2D full FOV MPI cannot detect signal from 6400 cells. Note the impeding gastrointestinal signal that is labeled "gut". (**B**) A BLI image of the same mouse showing detectable signal from the 6400-cell injection. This image shows BLI signal overlayed with X-ray. (**C**) 2D full FOV MPI could detect signals from 12,800 cells with impeding gut signals (labeled). (**D**) BLI signal was detected in the same mouse.

To assess the sensitivity of BLI for mice with hair, one C57Bl/6 mouse with black hair and one Balb/c mouse with white hair were both imaged after an injection of 12,800 labeled cells (Figure 7). White fur permitted the visualization of signal ($1.27 \times 10^5$ p/s/cm$^2$/sr) (Figure 7A); However, it doubled immediately after hair removal ($2.41 \times 10^5$ p/s/cm$^2$/sr) (Figure 7B). The signal continued to peak after hair removal ($4.02 \times 10^5$ p/s/cm$^2$/sr) (Figure 7C). BLI signal was completely attenuated by the black mouse hair on a C57Bl/6 mouse (Figure 7D). After shaving, the BLI signal was visible and measured $1.40 \times 10^5$ p/s/cm$^2$/sr (Figure 7E). The signal similarly continued to peak after hair removal ($8.36 \times 10^5$ p/s/cm2/sr) (Figure 7F).

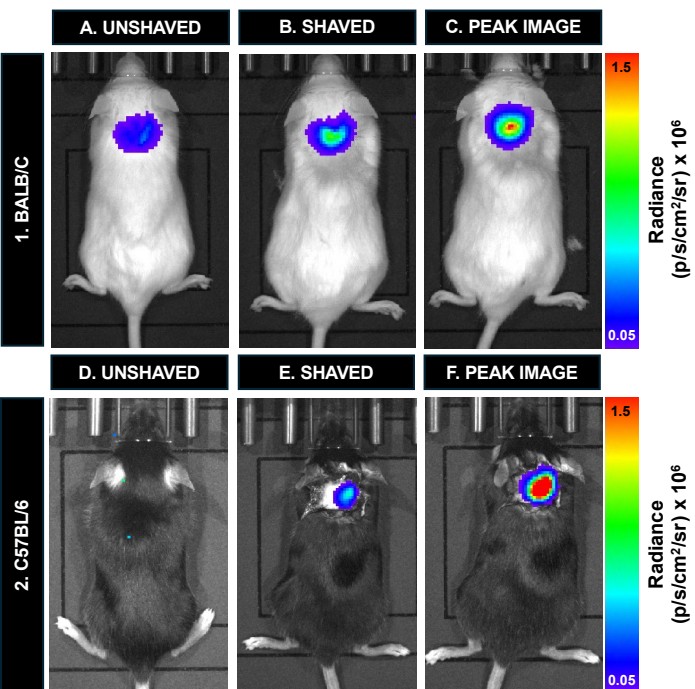

**Figure 7.** BLI cell detection in vivo for mice before versus after hair removal: (**A**) BLI signal is observed in the white mouse with hair. (**B**) Hair removal reveals an immediate increase in signal. (**C**) The signal continued to increase after hair removal. (**D**) Complete signal attenuation was observed for a C57Bl/6 mouse imaged with black hair. (**E**) BLI signal was observed with hair removal and (**F**) continued to increase until reaching peak radiance.

To compare the effect of depth on the sensitivity of MPI and BLI, we compared the signals from mice injected with 12,800 cells and imaged in the prone versus supine positions. The mouse's neck depth measured 2.5 cm. Figure 8A,B shows that the BLI signal was detected in the prone position but is completely attenuated in the supine position. Figure 8C,D shows that the MPI signal could be detected in both positions. The MPI signal, iron content, and cell number were all similar between prone and supine images. MPI signal was 201 A.U. in prone and 184 A.U. in supine. The iron content measured 0.126 μg in prone and 0.115 μg in supine. Finally, the calculated cell number was 7754 cells in prone and 7100 cells in supine.

The tables below list the BLI signal quantification (Table 2) and the MPI signal quantification (Table 3) values for each mouse imaged in this study. These values can be compared with the cell samples imaged previously with both BLI (Figure 3) and MPI (Figure 4). There is a reduction in signal for BLI and MPI when cells are imaged in vivo.

For 12,800 cells, there was an approximate 55% decrease in BLI signal in vivo ($n$ = 3 mice: nu/nu, Balb/c, and C57Bl/6) compared with cell samples ($n$ = 3) on average. Samples of 12,800 cells imaged in vitro had an average signal of $1.28 \times 10^6$ p/s/cm$^2$/sr, and mice imaged with 12,800 cells had an average signal of $5.72 \times 10^5$ p/s/cm$^2$/sr. Similarly, the in vivo MPI signal for 12,800 cells ($n$ = 3 mice: nu/nu, Balb/c, and C57Bl/6) decreased by approximately 49% on average compared with the cell samples ($n$ = 3) (average of 389 A.U. for cell samples versus 200 A.U. for mice). Cell samples were analyzed using the half-maximum method for this comparison. This MPI signal decrease affected the accuracy of cell number quantification in vivo, and the calculated cell number was 35–46% lower than 12,800 (actual cell number injected). Appendix A shows the MPI image of the C57Bl/6 mouse injected with 12,800 cells (Figure A2).

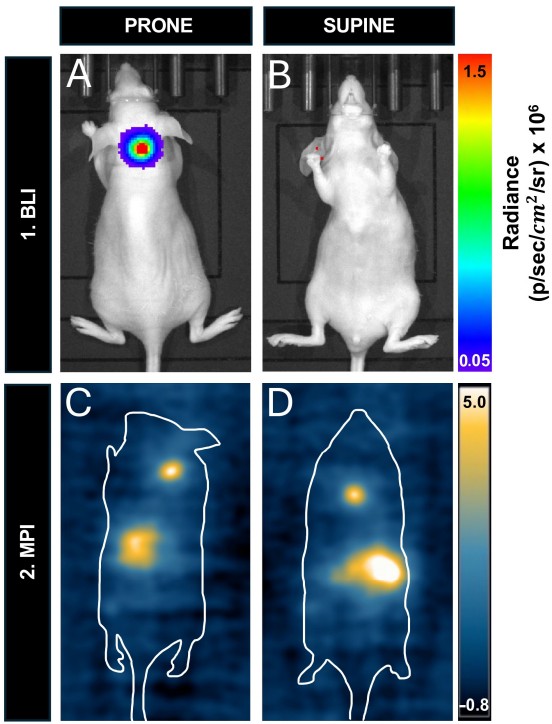

**Figure 8.** In vivo cell detection and sensitivity with prone and supine imaging for BLI and MPI: (**A**) In the prone position, signal is detected from 12,800 cells in the nude mouse. (**B**) This signal is completely attenuated when the mouse is imaged supine (depth of cell injection site = 2.5 cm). A Balb/c mouse containing 12,800 cells had a similar MPI signal when imaged in (**C**) prone and (**D**) supine.

**Table 2.** Quantification of BLI radiance ($p/s/cm^2/sr$) for all mice imaged in this study.

| Mouse | Figure | Number of Cells Injected | Imaging Position | Radiance ($p/s/cm^2/sr$) Before Hair Removal | Radiance ($p/s/cm^2/sr$) After Hair Removal | Peak Radiance ($p/s/cm^2/sr$) |
|---|---|---|---|---|---|---|
| Nu/nu | Figure 6B | 6400 | Prone | N/A | N/A | $6.38 \times 10^4$ |
| Nu/nu | Figure 6D | 12,800 | Prone | N/A | N/A | $4.77 \times 10^5$ |
| Balb/c | Figure 7A–C | 12,800 | Prone | $1.27 \times 10^5$ | $2.41 \times 10^5$ | $4.02 \times 10^5$ |
| C57Bl/6 | Figure 7D–F | 12,800 | Prone | Undetectable | $1.11 \times 10^5$ | $8.36 \times 10^5$ |
| Nu/nu | Figure 8B | 12,800 | Supine | N/A | N/A | Undetectable |

**Table 3.** Quantification of MPI signal, iron content, and cell number for all mice imaged in this study.

| Mouse | Figure | Number of Cells Injected | Imaging Position | Hair | Total Signal (A.U.) | Iron Content (μg) | Calculated Cell Number |
|---|---|---|---|---|---|---|---|
| Nu/nu | Figure 6A | 6400 | Prone | Hairless mouse | Undetectable | N/A | N/A |
| Nu/nu | Figure 6C | 12,800 | Prone | Hairless mouse | 217 | 0.135 | 8355 |
| Balb/c | Figure 8C | 12,800 | Prone | With hair | 201 | 0.126 | 7754 |
| Balb/c | Figure 8D | 12,800 | Supine | With hair | 184 | 0.115 | 7100 |

## 4. Discussion

In this study, we compared the sensitivity of MPI and BLI for imaging labeled cells. As expected, with BLI, we could clearly detect and quantify the lowest cell number studied, which was 100 cells in vitro and 6400 cells in vivo. These were firefly luciferase-expressing cancer cells. We did not image lower cell numbers with BLI; however, other studies

using fluc-expressing 4T1 cells have reported the detection of as few as two circulating 4T1 tumor cells in mouse blood samples [43] and as few as three 4T1 cells implanted subcutaneously [44]. The sensitivity of BLI can be improved through alterations to the luciferase—luciferin system to increase the number of photons produced by the reaction. Evans et al. developed an alternative luciferin substrate with a higher affinity for firefly luciferase, which resulted in a greater than 10-fold increase in signal intensity in vivo compared with d-luciferin at the same dose [45]. Hall et al. developed NanoLuc with a 150-fold increase in enzyme-specific activity compared with fluc [46]. Maximizing luciferase transcription and/or translation can also increase signal intensity. We employed this strategy by using FACS to select 4T1 cells with the highest expression of the luciferase–GFP construct.

With MPI, the lowest cell number detected in vitro was 800 4T1 cells using 3D MPI and 3200 cells using 2D MPI. Comprehensive reporting of MPI cell detection limits should include the imaging parameters, cell number, total amount of iron, iron/cell, SPIO, and cell type. The MPI signal will be different for the same cell type labeled with different SPIOs because some SPIOs produce more signal per mass of iron than others and because the amount of iron loading per cell will differ with different SPIO formulations. The MPI signal will also differ for different cell types labeled with the same type of SPIO, because some cells take up more particles and because the way cells compartmentalize particles can impact the MPI signal [30].

The 3D detection limit for ProMag-labeled 4T1 cells corresponds to ~13 ng of iron or 800 cells labeled with ~16 pg iron/cell, and the 2D detection limit corresponds to ~52 ng of iron or of 3200 cells labeled with ~16 pg with an SPIO called Synomag-D, the lowest mass of iron detected in vitro with 2D MPI was 28 ng, which corresponded to 8000 cells labeled with ~3.5 pg of iron/cell [5]. In this case, even though a lower amount of iron could be detected, the cellular detection limit was worse for Synomag-D. The higher sensitivity in terms of cell number for ProMag-labeled 4T1 cells is partly related to the more efficient cell labeling (3.5 pg/cell for Synomag-D versus 16 pg for ProMag). For example, if the cell loading for Synomag-D was 16 pg/cell (like for ProMag), 28 ng would represent 1750 cells, which would be a better cell detection limit compared with ProMag. This is without considering the magnetic properties of Synomag-D compared with ProMag. MPI relaxometry has shown that the peak MPI signal for free Synomag-D is ~3 times higher than ProMag. However, the MPI signal for Synomag-D has been shown to decrease by ~3 times after cell internalization, whereas the signal is unchanged for free ProMag versus intracellular ProMag, resulting in the same peak MPI signals for intracellular ProMag and Synomag-D [47].

In another study from our lab, 4T1 cells were labeled with a SPIO called Vivotrax [41]. Vivotrax is a ferucarbotran and has been widely used for MPI, even though it has a relatively low peak MPI signal compared with other SPIOs (Synomag-D, MPIO) [6,48]. The 3D detection limit for Vivotrax labeled 4T1 cells was 4000 cells, which corresponded to 36 ng of iron (~9.2 pg iron/cell) and 8000 cells with 2D MPI, corresponding to 74 ng of iron. The cellular detection limit was improved for mesenchymal stem cells (MSCs) labeled with Vivotrax. Here, the 3D detection limit for Vivotrax labeled MSCs was 2000 cells, which corresponded to 38 ng of iron (~19 pg iron/cell) and 4000 cells with 2D MPI, corresponding to 76 ng of iron. The enhanced Vivotrax labeling of MSCs is due to the higher phagocytic capacity and larger size of these cells [41].

In vivo mouse imaging revealed important limitations of both MPI and BLI. The main limitations of in vivo MPI were the low dynamic range and the quantification of cell number. The dynamic range of MPI is limited when multiple samples or signals with different iron concentrations are present in the FOV [42]. In this study, background signal from the gastrointestinal region obscured the lower signal of interest from the 6400 cells. We have previously shown that images of mouse chow produce MPI signal and that fasting mice can reduce but not eliminate this unwanted signal [4]. Doubling the cell number to 12,800 in another mouse allowed for the detection of signal despite the presence of gut signal. Quantification of the MPI signal, and conversion to iron mass and cell number, was

accurate for in vitro imaging of cell samples. However, there was a discrepancy between the number of cells injected into mice and the cell number calculated from the in vivo images. The MPI signal from 12,800 cells injected subcutaneously in vivo was reduced by ~50% compared with the 12,800-cell sample. This agrees with a previous study by our group, which found that for mice injected subcutaneously with 100,000 stem cells, the in vivo MPI signal was reduced by ~50% compared with cell pellets [41]. This is likely due to the dispersion of cells from the injection site, which reduces the number of cells per voxel, leading some cells to fall below the intravoxel detection limit. Particle immobilization in tissue may also contribute to the observed decrease in MPI signal, as reported by Arami et al. for SPIOs in agarose gel mimicking biological tissue [49]. Both of these limitations contribute to the lower cellular detection limit for 2D in vivo MPI (12,800 cells) compared with 2D in vitro MPI (3200 cells).

The main limitations of in vivo BLI were related to weak signal penetration, signal attenuation, and scattering. In our study, black mouse hair completely attenuated the BLI signal from 12,800 cells. This phenomenon has been observed by other groups. Puaux et al. showed that C57Bl/6 mice bearing melanoma tumors demonstrated a 74-fold decrease in BLI signal intensity when they were imaged unshaved, compared with after being shaved [37]. Curtis et al. further reported that after shaving, changes in skin pigmentation over the hair cycle for C57Bl/6 mice also significantly influenced the BLI signal intensity for the same mice imaged over multiple days [50]. For longitudinal cell tracking experiments involving multiple rounds of shaving and skin pigment changes, these drastic changes in BLI signal intensity can affect the validity and interpretation of results. We also found that white mouse hair from a Balb/c mouse caused partial signal attenuation and light scattering. Our results agree with a technical report by Tseng et al. [51] where a tumor-bearing Balb/c mouse showed a 50% increase in BLI signal intensity after hair removal. As has been demonstrated before, mouse hair can interfere with accurate BLI signal measurement, and it should be removed prior to imaging if possible. The challenge of imaging cells in deep tissues is well known for BLI. We demonstrated this by imaging a mouse in both the prone and supine positions. BLI signal could only be detected in the prone position. In the supine position, the cells were approximately 2.5 cm deep. There is no attenuation of MPI signal from biological tissue. Signal penetration can be improved for BLI by using a luciferase with a longer peak emission wavelength in the near-infrared spectrum. Akaluc, along with its substrate akalumine-HCl, is an analog system to fluc and d-luciferin. The akaluc system emits light at a longer wavelength (650 nm) than fluc (600 nm) and can outperform fluc [22,52]. Fluc can also be combined with the akalumine-HCl substrate for improved peak emission wavelength (670 nm) [53] and depth penetration [54], compared with the traditional D-luciferin substrate. However, hepatic background signal and skin toxicity have been reported in mice when using the akalumine-HCl substrate [53]. New luciferase and luciferin pairs are being developed with near-infrared emission spectra to improve the sensitivity and depth penetration of BLI [21].

Although not observed in this study, masking of signal can also occur with BLI. A large primary signal can saturate the CCD camera and prevent the visualization of a dim signal in the same animal. Williams et al. used akaluc BLI and MPI to monitor tumor growth and metastasis. Both modalities detected metastases resulting from a primary mammary fat pad tumor in nude mice; however, they were visualized in different anatomical regions. BLI detected metastases in the upper body, but this required the primary tumor to be covered, without this, the strong primary signal masks other signals in the mice. MPI signal was not detected in these regions, presumably due to the dilution of tracer with the proliferation of cancer cells, which reduces the amount of iron in the metastatic cells. However, MPI signal was detected in regions closer to the primary tumor, which were obscured in BLI [5]. The discrepancy in locating metastases with MPI and BLI highlights the complementary use of both modalities to give a more complete picture of cell fate.

Like MPI, the BLI signal was also reduced in vivo. For BLI this may be related to access to the luciferin substrates administered intraperitoneally. Cell samples have

immediate access to luciferin substrates in vitro, whereas in vivo, luciferin administered intraperitoneally diffuses into tissues and is then delivered through the circulation. Berger et al. found an uneven biodistribution of d-luciferin after intraperitoneal injection, with accumulation in the organs of the gastrointestinal system, the kidneys, and liver after 30 min [55]. Evans et al. also compared d-luciferin with a synthetic substrate, CycLuc1, which had a more uniform biodistribution after an intraperitoneal injection and remained in the mouse circulation for longer [45]. With a longer circulation time, more substrate would be delivered to luciferase-expressing cells, thereby increasing the BLI signal intensity. The BLI signal may also be reduced because of cell death, which can be caused by the subcutaneous injection. Ultimately, this is not as significant a limitation as it is for MPI, which promises direct quantification of iron content to estimate cell number.

The small number of animals used for in vivo imaging represents a limitation of this study. Additionally, we did not explore longer wavelength luciferase systems, such as akaluc, in this study, which may display better signal penetration in vivo. Future studies should image a larger cohort of animals and compare signal attenuation in vivo using different luciferases.

## 5. Conclusions

This study explored the advantages and disadvantages of MPI and BLI for imaging cells. The major advantages of MPI over BLI were the ability to quantify the iron mass from the MPI signal and the lack of signal attenuation by hair and tissues. However, the promise of accurate quantification of cell number from in vivo MPI requires further understanding of changes in the signal due to cell dispersion and the effects of the in vivo environment on the magnetic response of SPIOs, as well as advances in image acquisition and reconstruction methods. The major advantage of BLI over MPI was higher cellular sensitivity. By understanding how these two modalities complement each other and their individual limitations in sensitivity and cell detection, they can be more efficiently used together for cell tracking.

**Author Contributions:** Conceptualization, P.J.F. and S.T.; methodology, P.J.F. and S.T.; software, S.T.; validation, S.T.; formal analysis, S.T.; investigation, S.T. and B.N.; resources, P.J.F.; data curation, S.T.; writing—original draft preparation, S.T.; writing—review and editing, P.J.F., S.T. and B.N.; visualization, P.J.F. and S.T.; supervision, P.J.F.; project administration, S.T.; funding acquisition, P.J.F. All authors have read and agreed to the published version of the manuscript.

**Funding:** This research was funded by the Canadian Institutes of Health Research, grant number PJT 178223, and the National Sciences and Engineering Research Council of Canada, grant number 06671-2020.

**Institutional Review Board Statement:** The study was conducted in accordance with the Declaration of Helsinki and approved by the Animal Use Subcommittee of Western University's Council on Animal Care (AUP # 2023-113, Approved on 1 May 2024).

**Informed Consent Statement:** Not applicable.

**Data Availability Statement:** The data presented in this study are available upon request from the corresponding author.

**Acknowledgments:** We would like to acknowledge John J. Kelly and John A. Ronald for providing expertise to guide this project. We would also like to acknowledge the London Regional Flow Cytometry Facility for support and facility use to conduct flow cytometry experiments. We would also like to acknowledge the Imaging Pathogens for Knowledge Translation (ImPaKT) facility at Western University and the Canadian Foundation for Innovation.

**Conflicts of Interest:** The authors declare no conflicts of interest. The funders had no role in the design of the study; in the collection, analyses, or interpretation of data; in the writing of the manuscript; or in the decision to publish the results.

## Appendix A

**Table A1.** Cell counts before and after magnetic column separation (trial 1 of 3).

| Cell Count (Cells/mL) Before Magnetic Separation | Cell Count (Cells/mL) After Magnetic Separation |
|---|---|
| $4.75 \times 10^6$ | $4.70 \times 10^6$ |
| $5.78 \times 10^6$ | $4.61 \times 10^6$ |
| $3.80 \times 10^6$ | $3.74 \times 10^6$ |

**Table A2.** Cell counts before and after magnetic column separation (trial 2 of 3).

| Cell Count (Cells/mL) Before Magnetic Separation | Cell Count (Cells/mL) After Magnetic Separation |
|---|---|
| $7.87 \times 10^6$ | $7.01 \times 10^6$ |
| $7.46 \times 10^6$ | $6.61 \times 10^6$ |
| $6.72 \times 10^6$ | $6.24 \times 10^6$ |

**Table A3.** Cell counts before and after magnetic column separation (trial 3 of 3).

| Cell Count (Cells/mL) Before Magnetic Separation | Cell Count (Cells/mL) After Magnetic Separation |
|---|---|
| $4.82 \times 10^6$ | $4.91 \times 10^6$ |
| $5.88 \times 10^6$ | $5.24 \times 10^6$ |
| $5.88 \times 10^6$ | $5.65 \times 10^6$ |

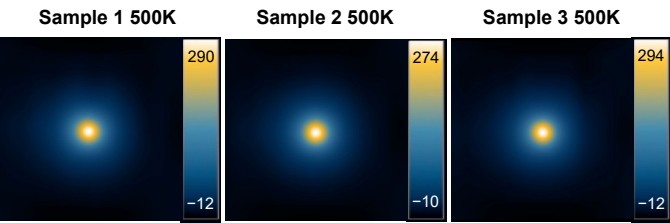

**Figure A1.** MPI images of triplicate 500,000 cell samples used to calculate the average iron loading per cell.

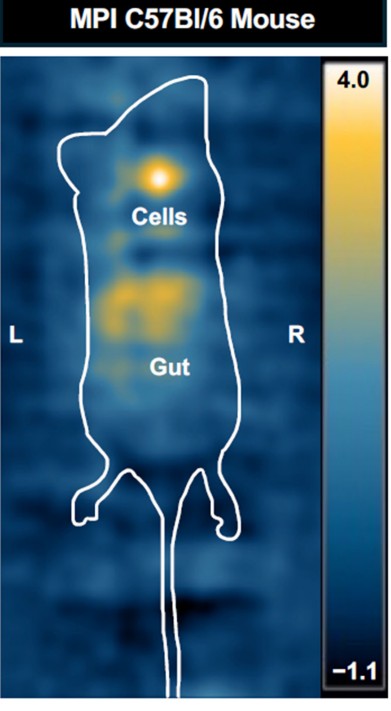

**Figure A2.** MPI image of C57Bl/6 mouse injected with 12,800 cells. The mouse was imaged with a full FOV in the prone position. The MPI signal (181 A.U.), iron content (0.113 μg), and cell number (6962 cells) were quantified for this mouse.

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
