# Peer review of "A Comparison of the Sensitivity and Cellular Detection Capabilities of Magnetic Particle Imaging and Bioluminescence Imaging"

_tomography, doi:10.3390/tomography10110135_

Round 1

Reviewer 1 Report

Comments and Suggestions for Authors

This manuscript is very well thought through and may be of great interest to the molecular imaging readership of tomography. All figures and methods are exhaustive and compelling. Two minor suggestions to improve the manuscript are as follows: 1) Add references of work by Bulte et al, Daldrup-link et al and Chapelin et al. for cell tracking applications. 2) The comparisons of BLI and ProMag vs Sinomag LOD was of great interest and adding a brief comparison to MRI detection limits with MPIOs and USPIOs would be very interesting and give more perspective to the reader as to the advantage of MPI over MRI.

Author Response

Comment 1: Add references of work by Bulte et al, Daldrup-link et al and Chapelin et al. for cell tracking applications.

Response: Thank you for the suggestion. We have added references to two comprehensive review articles on cell tracking from the Bulte, Ahrens and Daldrup-Link labs to the first sentence of the Introduction (page 1, paragraph 1, line 28). We have not added Chapelin papers because we have been asked to reduce the number of references and the Chapelin papers focus on 19F MRI which is of less relevance for this paper.

Comment 2: The comparisons of BLI and ProMag vs Sinomag LOD was of great interest and adding a brief comparison to MRI detection limits with MPIOs and USPIOs would be very interesting and give more perspective to the reader as to the advantage of MPI over MRI.

Response: Because the editor has asked that we shorten the Introduction and Discussion sections we cannot add text which compares MPI and MRI detection limits. However, we feel that this is covered in Table 1 (page 3).

Reviewer 2 Report

Comments and Suggestions for Authors

The manuscript by Trozzo et al. presents a comparison of the sensitivity and cellular detection capabilities of magnetic particle imaging (MPI) and bioluminescence imaging (BLI). The manuscript is well-written, with comprehensive control experiments, making it easy for the readership of Tomography to follow. It provides valuable information about BLI and MPI techniques. The authors lead with preliminary experimental results, effectively highlighting the limitations of each imaging technique. However, the manuscript does not conclude with a proposed solution to address these drawbacks. I have a few concerns that the authors should clarify or address through additional control experiments, if feasible, before the manuscript can be considered for publication.

  1.  In the supine position, there appears to be a reduction in signal with increased tissue depth. Did the authors explore any alternative techniques, such as using longer-wavelength luciferins or enhanced optical systems, to mitigate the depth-related limitations in BLI?
  2.  In in vitro experiments, the authors observed a strong correlation between BLI and MPI signals across a range of cell numbers. Did the authors detect any differences in the dynamic range of these modalities at lower cell concentrations, particularly below 1,000 cells?
  3. How does ProMag labeling impact long-term cell viability in vivo, beyond the initial assessments? Have the authors observed any delayed cytotoxic effects in labeled cells?
  4. Is there a specific reason the authors chose to use the 4T1 cell line? For broader applicability, the authors should consider conducting a control experiment using a different cell line to generalize the findings.
  5. For in vivo imaging, the authors reports reduced BLI signal intensity. Could the authors provide more details or clarify this observation?
  6. The authors should provide a citation for the statement made in lines 82-84 to support the claim.

Author Response

Comment 1: In the supine position, there appears to be a reduction in signal with increased tissue depth. Did the authors explore any alternative techniques, such as using longer-wavelength luciferins or enhanced optical systems, to mitigate the depth-related limitations in BLI?

Response: We acknowledge that longer wavelength luciferases such as akaluc can improve BLI sensitivity, specifically mitigating depth-related limitations and have noted this in the Discussion section (paragraph 1, page 16). We did not explore alternative systems in this study.

We selected firefly luciferase (fluc) for this study because it is the most commonly used system. With this decision, we aim for our results to be applicable to the broader research community. As discussed in the manuscript, akaluc can produce background signals in the liver1 and the akalumine-HCl substrate has been associated with skin reactions in mice1,2 (page 17, paragraph 3). Newly developed alternative, red-shifted luciferases are available and can be explored in future studies. In the discussion section of the manuscript, we highlight that longer wavelength luciferases are available and that they could improve depth penetration. We also provide a brief comparison of fluc and akaluc for imaging.

  1. Amadeo, F., Plagge, A., Chacko, A., Wilm, B., Hanson, V., Liptrott, N., Murray, P., Taylor, A. (2021) Firefly luciferase offers superior performance to AkaLuc for tracking the fate of administered cell therapies, Eur. J. Nucl. Med. Mol. Imaging, DOI: 10.1007/s00259-021-05439-4 
  2. Nakayama, J.; Saito, R.; Hayashi, Y.; Kitada, N.; Tamaki, S.; Han, Y.; Semba, K.; Maki, S. A. High Sensitivity In Vivo Imaging of Cancer Metastasis Using a Near-Infrared Luciferin Analogue seMpai. Int. J. Mol. Sci.202021, 7896,  DOI: 10.3390/ijms21217896 

Comment 2: In in vitro experiments, the authors observed a strong correlation between BLI and MPI signals across a range of cell numbers. Did the authors detect any differences in the dynamic range of these modalities at lower cell concentrations, particularly below 1,000 cells?

Response: In vitro, the lowest number of cells that could be detected and quantified for 2D MPI was 3,200 cells. We did not directly compare the BLI and MPI signals for cell numbers lower than this.

Comment 3: How does ProMag labeling impact long-term cell viability in vivo, beyond the initial assessments? Have the authors observed any delayed cytotoxic effects in labeled cells?

Response: We agree that it is important to understand the long-term effects of iron-labeling on cell viability. To address this, we conducted an additional in vitro experiment to compare the BLI signal of iron-labeled and unlabeled 4T1 cells over a period of five (5) days. BLI signal is only produced by live cells and can therefore serve as a measure of cell viability. We observed no significant differences in BLI signal between labeled and unlabeled cells over the course of the 5-day period. These findings are similar to a previous paper by our group which assessed the BLI signal of Synomag-D labeled 4T1 cells compared to unlabeled cells over a 7-day period1.

The revised manuscript has been edited to incorporate this additional experiment. Specifically, we have altered Figure 1 (page 8). Previously, Figure 1E and 1F depicted the BLI signal of iron labeled and unlabeled 4T1 cells imaged only after 24-hours of labeling. These figures have been replaced by the new experiment, which compares the BLI signal over a 5-day period. Figure 1E shows representative BLI images of the labeled and unlabeled cells on day 1 and day 5 of imaging. Figure 1F shows the corresponding BLI signal measurements for each day of imaging. We have also revised section 2.5.1 of the methods (page 5, paragraph 4, lines 385-396) to reflect the additional procedures related to this experiment.

  1. Williams, R.J.; Sehl, O.C.; Gevaert, J.J.; Liu, S.; Kelly, J.J.; Foster, P.J.; Ronald, J.A. Dual Magnetic Particle Imaging and Akaluc Bioluminescence Imaging for Tracking Cancer Cell Metastasis. Tomography 2023, 9, 178–194, doi:10.3390/tomography9010016.

Comment 4: Is there a specific reason the authors chose to use the 4T1 cell line? For broader applicability, the authors should consider conducting a control experiment using a different cell line to generalize the findings.

Response: We chose to use the 4T1 cell line because we have used these cells in the past for several other cancer cell tracking studies. In Sehl and Foster, Scientific Reports, 2021 (https://doi.org/10.1038/s41598-021-01642-3) we assessed the cellular sensitivity of MPI for the detection of 4T1 cells labeled with a different SPIO, a ferucarbotran called Vivotrax which is the most commonly used SPIO for MPI.  We were interested in comparing the results of that study with ProMag labeled 4T1 cells.  

We appreciate that our detection limits would be different with other cell types; some cells will take up more iron than others and this will lead to the detection of fewer cells. We have previously reported on this. In the paper mentioned above the detection limits for Vivotrax labeled 4T1 cells were compared with Vivotrax labeled stem cells. as few as 4000 stem cells (76 ng iron) and 8000 breast cancer cells (74 ng iron) were reliably detected with MPI. In paragraphs 3-4 of the Discussion (page 16-17), we provide a comparison of different cell detection limits observed for MPI using different cell lines with different iron loading efficiency.  

Comment 5: For in vivo imaging, the authors reports reduced BLI signal intensity. Could the authors provide more details or clarify this observation?

Response: We compared the peak BLI signal for 12,800 cells imaged in vitro (n=3) and in vivo (n=3). Three samples of 12,800 cells were imaged in vitro and peak radiance values are reported in Figure 3. Three mice (one C57Bl/6, one nu/nu, and one Balb/c) were injected with 12,800 cells for in vivo BLI imaging. The peak radiance values for these mice (after shaving for Balb/c and C57Bl/6 mice) are reported in Table 2.

In vitro, the peak radiance for 12,800 cell samples (n=3) was 1.28 x 106 p/sec/cm2/sr on average. In vivo, the peak radiance for 12,800 cell samples (n=3) was 5.72 x 105 p/sec/cm2/sr on average. To determine the % signal decrease from in vitro to in vivo imaging, we used the following calculation:  

% signal decrease (average) =  1 - (5.72 x 105 / 1.28 x 106) = 55%

We clarify this result on lines 715-717 of the results (page 15, paragraph 2) in the revised manuscript. 

Comment 6: The authors should provide a citation for the statement made in lines 82-84 to support the claim.

Response: To further support the discussion of MPI physics, we have added a sentence to this section of the introduction which directly refers readers to two MPI review papers which provide more detailed information: Harvell-Smith et al1 and Chandrasekharan et al2. This can be found on page 2, paragraph 3, lines 94-96. 

  1. Harvell-Smith S, Tung LD, Thanh NTK. Magnetic particle imaging: tracer development and the biomedical applications of a radiation-free, sensitive, and quantitative imaging modality. Nanoscale. 2022 Mar 10;14(10):3658-3697. doi: 10.1039/d1nr05670k. PMID: 35080544
  2. Chandrasekharan, P.; Tay, Z.W.; Zhou, X.Y.; Yu, E.; Orendorff, R.; Hensley, D.; Huynh, Q.; Fung, K.L.B.; VanHook, C.C.; Goodwill, P.; et al. A Perspective on a Rapid and Radiation-Free Tracer Imaging Modality, Magnetic Particle Imaging, with Promise for Clinical Translation. Br J Radiol 2018, 91, 20180326, doi:10.1259/bjr.20180326.

Round 2

Reviewer 2 Report

Comments and Suggestions for Authors

Authors have now revise/explained all the concerns. I have no further commments.